# X-ray Photoelectron Spectroscopy Analysis of Scandia-Ceria-Stabilized Zirconia Composites with Different Transport Properties

**DOI:** 10.3390/ma16165504

**Published:** 2023-08-08

**Authors:** Iraida N. Demchenko, Kostiantyn Nikiforow, Maryna Chernyshova, Yevgen Melikhov, Yevgen Syryanyy, Nadiia Korsunska, Larysa Khomenkova, Yehor Brodnikovskyi, Dmytro Brodnikovskyi

**Affiliations:** 1Institute of Plasma Physics and Laser Microfusion, ul. Hery 23, 01-497 Warsaw, Poland; iraida.demchenko@ifpilm.pl (I.N.D.); maryna.chernyshova@ifpilm.pl (M.C.); ysyryanyy@gmail.com (Y.S.); 2Institute of Physical Chemistry Polish Academy of Sciences, ul. Kasprzaka 44/52, 01-224 Warsaw, Poland; knikiforow@ichf.edu.pl; 3Institute of Fundamental Technological Research Polish Academy of Sciences, ul. Pawinskiego 5b, 02-106 Warsaw, Poland; 4Institute of Microelectronics and Optoelectronics, Warsaw University of Technology, Koszykowa 75, 00-662 Warsaw, Poland; 5V. Lashkaryov Institute of Semiconductor Physics, National Academy of Sciences of Ukraine, 45 Nauky Ave., 03028 Kyiv, Ukraine; kors@isp.kiev.ua (N.K.); khomen@isp.kiev.ua (L.K.); 6Frantsevich Institute for Problems of Materials Science, NAS of Ukraine, Krzhizhanovskoho Str. 3, 03142 Kyiv, Ukraine; bregor@ipms.kiev.ua (Y.B.); brodnikovskii.d@nas.gov.ua (D.B.)

**Keywords:** XPS, zirconia, scandia-ceria-stabilized zirconia, ScCSZ, SOFC

## Abstract

This work aims to study a possible modification in the electronic structure of scandia-ceria-stabilized zirconia (10Sc1CeSZ) ceramics sintered at different temperatures. In addition to using X-ray diffraction (XRD), scanning electron microscopy (SEM) and impedance spectroscopy to investigate the structural and electrical properties, we employed X-ray photoelectron spectroscopy (XPS) to determine the chemical state information of the atoms involved, along with compositional analysis. As expected, a significant increase in grain ionic conductivity with the sintering temperature was present. This increase was accompanied by a decrease in the porosity of the samples, an increase in the grain size, and a transformation from the rhombohedral to the cubic phase. The phase transformation was detected not only using XRD, but also using XPS and, for this type of ceramic, XPS detected this transformation for the first time. In addition to the changes in the structural characteristics, the increase in the ionic conductivity was accompanied by a modification in the electronic structure of the ceramic surface. The XPS results showed that the surface of the ceramic sintered at the lower temperature of 1100 °C had a higher amount of Zr–OH bonds than the surface of the ceramic sintered at the higher temperature of 1400 °C. The existence of these Zr–OH bonds was confirmed using Fourier-transform infrared spectroscopy (FTIR). From this result, taken together with the difference between the oxygen/zirconium ratios in these ceramics, also identified using XPS, we conclude that there were fewer oxygen vacancies in the ceramic sintered at the lower temperature. It is argued that these two factors, together with the changes in the structural characteristics, have a direct influence on the conductive properties of the studied ceramics sintered at different temperatures.

## 1. Introduction

Solid oxide fuel cell (SOFC) electrolytes continue to be actively researched and developed [1,2,3]. ZrO_2_ doped with Sc_2_O_3_ has proven to be a suitable material for the electrolyte because, in addition to good mechanical properties, it is characterized by the highest ionic conductivity among zirconia-based SOFC electrolytes [4,5,6,7,8,9,10,11,12,13,14,15]. This is due to the fact that a dopant ion should have an ionic radius as close as possible to the ionic radius of a host ion, i.e., zirconium. If this condition is fulfilled, it will reduce the steric-blocking effect during oxygen ion transport. Thus, ZrO_2_ doped with Sc_2_O_3_, being in the cubic phase, has a greater ionic conductivity than, for example, ZrO_2_ doped with Y_2_O_3_. Indeed, the ionic radius of Sc^3+^ is 0.89 Å, which is much closer to the value of 0.86 Å, for the ionic radius of Zr^4+^, than the value of 0.94 Å, for the doping ion Y^3+^ (note that the ionic radii of the elements are given with a six-fold coordination by oxygen atoms).

It is generally accepted that the polymorphic stability of zirconia is associated with the presence of oxygen vacancies in the lattice [16,17]. During the sintering of scandia and zirconia powders, trivalent scandium ions can replace zirconium (Zr^4+^) in the host lattice, with the formation of negatively charged defects, ${Sc}_{Zr}^{\prime }$, which, in turn, results in the creation of oxygen vacancies for charge compensation. The oxygen vacancies associated with the “relaxed” lattice stabilize the rhombohedral/cubic phase at room temperature, preventing spontaneous transformation into a tetragonal/monoclinic phase, and promote oxygen diffusion in ZrO_2_-based electrolytes [5,9].

It is known that, among the different phases of the Sc-stabilized zirconia (ScSZ) system, the highest ionic conductivity suitable for use as an SOFC electrolyte is achieved for a system in a cubic phase. However, in this system, the transition from the highly conductive cubic phase to the rhombohedral phase occurs at temperatures around 650 °C during cooling, or due to ageing at high temperatures (700–1000 °C), resulting in a sharp reduction in the electrical conductivity. As a result, this material is not suitable for use in low- or intermediate-temperature SOFCs. In order to stabilize the cubic phase of ScSZ at lower temperatures, various additional dopants, such as CeO_2_, Sm_2_O_3_, Yb_2_O_3_, and Al_2_O_3_, have been investigated [18,19,20,21]. Among them, the highest ionic conductivity was achieved for 1 mol% CeO_2_ and 10 mol% Sc_2_O_3_ added to ZrO_2_ (10Sc1CeSZ) [22]. However, the ionic conductivity of the 10Sc1CeSZ ceramic electrolyte depends on the powder characteristics, such as the particle size, surface area, and purity, which are affected by the preparation method, and by the purity of the raw materials [23,24].

On the other hand, the ionic conductivity is also significantly affected by the ceramic sintering regime, with the sintering temperature being one of the important parameters of this process. However, there are only a few works that have studied the effect of the sintering temperature on conductivity. Furthermore, different types of dependence of ionic conductivity on sintering temperature have been reported. For example, it was shown that an increase in the sintering temperature usually increased the conductivity [25,26]. However, sometimes, this dependence was non-monotonic [25]. The dependence of the ionic conductivity on the sintering temperature was explained by changes in the structural ceramic characteristics, such as the porosity, grain shape and size, phase composition, and grain boundary structure. In particular, the non-monotonic dependence of the ceramic conductivity on the sintering temperature was explained in [25] by the non-monotonic dependence of the porosity and non-uniform grain growth. At the same time, Brodnikovska et al. [27] showed that, in addition to the changes in conductivity, there is a dependence of the optical absorption on the sintering temperature, which was attributed to the changes in the content of point defects in the grain volume, such as impurities and oxygen vacancies. Thus, not only the structural characteristics of the ceramic, but also the defects present, have to be considered as a possible cause of the influence of the sintering temperature on the conductivity. However, to the best of our knowledge, this issue has rarely been addressed.

This study aimed to look into this question; i.e., to monitor the possible electronic structure modification of scandia-ceria-stabilized zirconia (ScCSZ) ceramics sintered at different temperatures. Standard measurements of the structural characteristics and ionic conductivity were carried out, but the main focus was on the use of X-ray photoelectron spectroscopy (XPS), which allows the unambiguous and highly accurate determination of chemical state information. The influence of the elemental composition and chemical environment (e.g., the oxidation state) of the sample surface on the transport properties of the samples is discussed.

## 2. Materials and Methods

Scandia-ceria-stabilized zirconia powder with a nominal composition of 89 mol% ZrO_2_-10 mol% Sc_2_O_3_-1 mol% CeO_2_ (10Sc1CeSZ) was obtained from Daiichi Kigenso Kagaku Kogyo (DKKK, Tokyo, Japan). For the sample preparation, the powders were milled with zirconia balls in ethyl alcohol for 24 h, and dried in air. The samples were pressed at 63 MPa into pellets of 15 mm × 1.5 − 2 mm, and then sintered at different temperatures within the interval of 1100–1400 °C, for 1.5 h, in air. Two samples, sintered at the temperatures of 1100 °C and 1400 °C, hereafter referred to as DKKK-1100 and DKKK-1400 respectively, were selected for the XPS study. The ionic conductivity measurements were carried out on the samples sintered at all temperatures, including the intermediate temperatures.

The X-ray diffraction (XRD) was measured in the Bragg–Brentano (Θ-2Θ) geometry, using a Philips X’Pert-MRD diffractometer (Amsterdam, The Netherlands) with CuKα1 radiation, at a wavelength of 0.15418 nm. The porosity of the sintered ceramic samples was determined via hydrostatic weighing. The grain sizes were evaluated via scanning electron microscopy (SEM), using an Auger microprobe JAMP-9500F (JEOL, Akishima, Japan).

The ionic conductivity was measured via impedance spectroscopy at temperatures of 400–900 °C in the frequency range of 10^−2^–10^6^ Hz, using the Solartron 1260 Impedance/Gain-Phase Analyzer (Denver, CO, USA) (for details see [28]). Silver paste electrodes were brushed onto the sample surfaces, and burned off at an appropriate temperature. The impedance spectra were modeled by means of two parallel equivalent R-CPE circuits connected in series. The low-frequency semicircle was attributed to the grain boundaries, while the high-frequency semicircle was assigned to the grains. The grain semicircle was visible on the impedance diagrams at measurements only up to 500 °C. At higher temperatures, it disappeared from the impedance diagrams.

The surface of the samples was investigated via XPS, using a PHI5000 VersaProbe spectrometer (Chanhassen, MN, USA), with monochromatic Al Kα radiation (hν = 1486.6 eV). The high-resolution (HR) XPS spectra were collected with the hemispherical analyzer, at a pass energy of 23.5 eV, an energy step size of 0.1 eV, and a photoelectron take-off angle of 45° with respect to the surface plane. Surface charge compensation was achieved using a low-energy electron flood gun. The photoelectron binding energies (BEs) were calibrated using the C-1s photoelectron peak at 284.8 eV. CasaXPS software, version 2.3.17dev6.6o [29] was used to deconvolute the XPS spectra.

Attempts to use ion beam sputtering to improve the surface quality of the prepared samples, i.e., to reduce the surface contamination, were unsuccessful, because such treatment leads to surface degradation, as discussed previously [30]. Therefore, in order to obtain a sample surface with as little contamination as possible, the samples were chipped (i.e., not cut), and the chipped surface was examined. Thus, although the XPS spectra reflect the properties of a surface layer, it is assumed here that the surface layer obtained via this technique is free from typical contaminants. In addition, the C-OH, O-C=O, and C=O groups revealed via the fitting procedure were formed naturally in an ambient environment, and did not influence the bulk properties. Therefore, they were excluded from further discussion.

## 3. Results and Discussion

The grain conductivity is one of the most important characteristics of fuel cell electrolytes, because oxygen diffusion through the grains makes an essential contribution to the electrolyte conductivity. The dependence of the grain conductivity of 10Sc1CeSZ ceramics measured at 500 °C on the sintering temperature is shown in Figure 1a, where it can be seen that the conductivity increases strongly with the increasing sintering temperature.

One of the reasons for the increase in grain conductivity with the sintering temperature could be a decrease in the ceramic porosity, caused by the decrease in the ceramic volume involved in conduction. It was found that the porosity does indeed decrease from the value of ~50%, for the sample sintered at 1200 °C, to the value of 1%, for the sample sintered at 1400 °C; see Figure 1b. However, it is unlikely that the decrease in the ceramic volume involved in conduction *alone* could explain such a large increase. In particular, the grain surface area, which is in contact with the atmosphere, also increases, leading to another effect of the porosity on the conductivity.

An important factor to consider is the increase in the grain size. The grain size increases from 100–200 nm to 500 nm, as the sintering temperature increases from 1100 to 1400 °C, respectively (see Figure 1b). An increase in the grain size can lead to two opposite trends: either an *increase* in the grain conductivity, due to a decrease in the grain boundary volume, or a *decrease* in the grain conductivity due to a decrease in the extended defect content. The first trend may be not very pronounced, due to the small grain boundary width (several nm) [31]. As for the second, its significance could be assessed from the coherent domain sizes obtained via XRD. In addition, XRD can also provide information on the phase composition, which, in turn, also affects the ionic conductivity. The XRD patterns of the samples sintered at different temperatures are shown in Figure 2.

As shown in Figure 2, for ceramics sintered at 1100 °C, two wings near the peaks at 30° (Figure 2b), 50°, and 60° (Figure 2c) are observed, indicating the presence of the rhombohedral phase. At a higher sintering temperature, only the cubic phase is registered. The estimation of the coherent domain size from the XRD peak width shows that its value changes slightly (from 22 nm to 32 nm), and should not lead to a significant increase in the conductivity.

In order to examine the other possible factors that may influence the difference in the ionic conductivity of samples sintered at different temperatures, the XPS study, which provides quantitative information on the elemental composition and chemical environment, was carried out.

In order to study the properties of ScCSZ in the context of the chemical states of the dopant and the host matrix ions, the BEs of a reference sample, i.e., standard cubic ZrO_2_, should be recalled. Basahel et al. [32] reported that Zr-3d_5/2_ and Zr-3d_3/2_ for cubic ZrO_2_ were found at 182.0 and 184.4 eV, respectively. For the DKKK-1400 sample, the values of the Zr-3d_5/2_ line and the spin–orbit coupling (182.17 ± 0.1 eV and 2.4 eV, respectively) are very close to the reference ones. A difference of 0.17 ± 0.1 eV between the values for cubic ZrO_2_ and for the DKKK-1400 ceramic could be explained by an error related to the calibration of the BE scale. Indeed, using a similar system, the annealed 9 mol% Y_2_O_3_-stabilized ZrO_2_ sintered at 1600 °C, which is in the cubic phase, Guo [33] found the BEs of 181.95 and 184.35 eV, for Zr-3d_5/2_ and Zr-3d_3/2_, respectively. In addition, Guo used the value of 284.6 eV from the C-1s photoelectron peak for calibration [33], which should be compared with our calibration value of 284.8 eV. The difference of ~0.2 eV between those values and ours is obvious.

However, for the DKKK-1100 sample (see Figure 3a), the BE of the Zr-3d_5/2_ line is 182.54 ± 0.1 eV, and is shifted by about 0.5 eV with respect to the cubic ZrO_2_ (the spin–orbit coupling value of 2.4 eV is the same, however). This value is closer to the one reported in [34]: a Zr-3d_5/2_ BE maximum standing at 182.4 ± 0.2 eV was reported for the stabilizing cerium oxide with yttrium, namely 25.5 wt.% CeO_2_-2.5 wt.% Y_2_O_3_-72 wt.% ZrO_2_, which forms a solid solution with zirconia in the cubic phase. One possibility to explain such a relatively large *positive* (+0.5 eV) shift of the Zr-3d_5/2_ line of the DKKK-1100 sample could be related to a chemical shift due to the high concentration of scandium on the surface of this sample, as seen in Table 1. Indeed, core-level chemical shifts could occur in the case of actual charge transfer between the constituent ions. As scandium and zirconium ions have unequal charges (+3 and +4, respectively), a charge redistribution may take place between them. If this is the case, it could only *reduce* the average positive charge of the zirconium, leading to a *lower BE* for Zr-3d levels with respect to the reference cubic ZrO_2_. Therefore, the discussed shift of the Zr-3d line cannot be explained by a chemical shift.

Another reason for the shift of about 0.5 eV of the Zr-3d_5/2_ BE to exist could be a change in the Madelung potential. Indeed, in addition to the charge redistribution, the presence of scandium/cerium ions at zirconium sites could affect the Madelung potential significantly, and thus could influence the core level BEs. The results of the XRD are in support of this hypothesis. The analysis of the spectra clearly identified that, instead of the presence of only a single cubic phase of ScCSZ, the ceramic sintered at 1100 °C consists of a mixture of a cubic phase and a small amount of secondary rhombohedral phase. All ScCSZ ceramics sintered at 1300 °C or higher temperatures have a cubic phase at room temperature. This observation is in good agreement with the results obtained via Rietveld analysis on similar composites [23]. Consequently, the XRD results allow the explanation of the shift in the XPS spectra of the Zr-3d_5/2_ line presented in Figure 3a, by the presence of a rhombohedral *β* phase at the chipped surface of the DKKK-1100 ceramic. Furthermore, the identification of this phase as a rhombohedral *β* phase, rather than other phases, seems reasonable: the BE values for the tetragonal ZrO_2_ phase would be at even higher energies, at 182.7 and 184.7 eV [36]. It should be noted that the existence of a *pure* cubic phase deeper in the grains cannot be excluded, even in a case where the samples are chipped, as the calculated “sampling depth” of photoelectrons registered by the analyzer in such a geometry is about 2.0–6.9 nm [37]. As noted in Section 2, the sputtering process (depth profiles) distorts the spectral profiles and, therefore, cannot be used here. However, the combination of XRD with XPS analysis allowed us to conclude that the core of the grains does have a cubic phase, while the surface reveals a rhombohedral phase. Thus, one of the reasons behind the lower conductivity of the DKKK-1100 sample is the presence of a low-conductive rhombohedral phase.

The analysis of the XPS spectra for scandium was straightforward, as the fitting with a doublet of appropriately constrained peaks is sufficient; see Figure 4a. The deconvoluted BE values of the Sc-2p_3/2_ line for the DKKK-1100 and DKKK-1400 samples are 402.54 ± 0.1 eV and 402.05 ± 0.1 eV, respectively. The values of the spin–orbit coupling are 4.49 and 4.45 eV for DKKK-1100 and DKKK-1400, respectively. The estimated BE value for DKKK-1400 is a clear indication of the presence of Sc^3+^ in the cubic phase of Sc_2_O_3_ [38,39,40]. The observed shift of the Sc-2p_3/2_ line of about 0.5 eV, and the higher spin–orbit coupling value for the DKKK-1100 ceramic could also be explained by the presence of the rhombohedral *β* phase. To the best of our knowledge, there are no data in the literature on the values of the BE of Sc^3+^ in rhombohedral (either *β* or *γ*) phases.

In any case, the atomic percentage of Sc^3+^ ions estimated by the deconvolution procedure is somewhat higher for the DKKK-1100 ceramic than for the DKKK-1400; see Table 1. In this respect, a higher number of oxygen vacancies for the sample sintered at the lower temperature can be expected. However, this is not the case. In fact, the analysis of the O-1s line (see Figure 4b), with its maximum at 530.35 ± 0.1 eV for DKKK-1100 and 529.92 ± 0.1 eV for DKKK-1400 (both values correspond to the Zr-O_LATT_ signal), allows us to conclude that the ratio of oxygen to zirconium in the ScCSZ ceramics is 1.8 and 1.5, respectively (see Table 1). Thus, the lower number of oxygen vacancies is obviously one of the reasons behind the lower conductivity of the sample sintered at 1100 °C.

In turn, the Zr–OH bond (with the BE at 531.5 ± 0.1 eV for both ScCSZ ceramics) content at the surface of the DKKK-1100 ceramic is about 1.9 times higher than in DKKK-1400 (Table 1). Consequently, the observed deterioration of the transport properties of the sample sintered at the lower sintering, DKKK-1100, compared to the sample sintered at a higher temperature, DKKK-1400, can be also associated with the penetration of ${OH}^{-}$ ions deeper inside via grain-boundary diffusion, and the consequent annihilation of oxygen vacancies. Indeed, assuming that the samples have been in the air atmosphere for some time, the source of ${OH}^{-}$ ions may be present from existing water molecules [33,41]. For a given number of ${OH}^{-}$ ions, they will contribute differently to the conductivity, depending on the grain surface area or the size of the pores. Qualitative analysis of the XPS spectra presented in Table 1 shows that the number of Zr–OH bonds is higher in the sample with the lower sintering temperature, DKKK-1100. According to the mechanism described in [33,41], the results from XPS could be explained as more ${OH}^{-}$ ions from the grain surface having penetrated the grain, and having deactivated a high number of the oxygen vacancies in it, leading to a negative effect on the transport properties of the DKKK-1100 ceramic. The opposite picture is observed for the ceramic sintered at a higher temperature, DKKK-1400. The obtained amount of zirconium bonded to OH in this ceramic is much lower compared to DKKK-1100. Through similar considerations, it can be concluded that the lower number of vacancies in this ceramic was deactivated by the ${OH}^{-}$ ions. The higher number of active vacancies in this sample leads to a better ionic conductivity in this sample. Such a mechanism was previously reported by Guo [33,41], and is in good agreement with the results presented here. To summarize, the “deactivation” of vacancies by the ${OH}^{-}$ ions that penetrate from the ceramic surface will lead to a greater or lesser influence on the final ionic conductivity of the material studied, as it directly affects the number of active oxygen vacancies. The existence of the OH bound to zirconium itself was confirmed via the Fourier-transform infrared (FTIR) measurements shown in Appendix A. The obtained FTIR results agree well with the data reported in [42].

The analysis of the Ce-3d spectra for both the examined ceramics can significantly contribute to our better understanding of the reasons behind the stabilization of ZrO_2_ ceramics doped with Sc_2_O_3_ via CeO_2_ doping. This is due to the fact that the reversible Ce(IV)O2-Ce(III)2O3 reduction transition, associated with the formation and migration of oxygen vacancies, is directly coupled with the process of the localization/delocalization of the Ce-4*f* electron [43]. The Ce-3d spectra for both the examined ceramics are shown in Figure 3b. The spectra have a complex structure, which is manifested by the appearance of six peaks in the spectrum of CeO_2_ (marked in green), and four peaks in that of Ce_2_O_3_ (marked in red). The complexity of the Ce-3d spectrum was described in detail by Kotani et al. [44], where it was shown that it arises from the proximity of the CeO2-4*f* level to the O-2*p* valence band with which it hybridizes. This requires the careful deconvolution of the resulting spectra. Among all the chemical and physical analytical instruments, the Ce-3d spectra show the best ability to observe the presence of Ce^4+^ and Ce^3+^ ions, and to determine their individual contents. The complexity of the Ce-3d line becomes even more pronounced when both 3+ and 4+ ions coexist at the surface, as at least ten peaks appear in the analyzed data, with some of them being too close to each other, requiring special deconvolution procedures to resolve the combination of the peak sets of both ions with the appropriate surroundings. It is worth emphasizing that the peak intensity ratios, which are determined by the electronic properties of a given cerium oxide, should be used as constraints when fitting the peak set of these particular ions with a particular surrounding, such as, e.g., in the oxides Ce_2_O_3_ or CeO_2_. An additional constraint that must also be considered is the inter-component intensity ratio associated with the j–j coupling, equal to 1.50, which applies to the electron population division into the 5/2 and 3/2 components of the Ce-3d level [45]. Here, the fitting procedure originally proposed by Pfau and Schierbaum [35], and subsequently tested by Paparazzo [45], was adopted. The main observation from Figure 3b and Table 1 is that Ce ions (3+,4+) coexist at the surface of the DKKK-1100 and DKKK-1400 ceramics, with the fraction of Ce^3+^/(Ce^3+^+Ce^4+^) equal to 0.59 and 0.61, respectively. Such a result makes it possible to assume that the number of oxygen vacancies due to the Ce(IV)O2-Ce(III)2O3 reduction transition is at the same level at the surface for both ceramics. The estimated Ce-3d_5/2_ BE values corresponding to 4+ and 3+ ions for the DKKK-1100 and DKKK-1400 ceramics are 882.5 ± 0.1 and 880.6 ± 0.1 eV, versus 882.13 ± 0.1 and 880.16 ± 0.1 eV, respectively. The noticeable shift of about 0.4 eV between the respective BE positions could also be explained by the difference in the ceramic phase at the sample surface (rhombohedral *β* versus cubic, the presence of which was confirmed through XRD studies). The estimated Ce-3d BE values are in good agreement with the published data [38,45,46,47,48].

## 4. Conclusions

In this work, 10Sc1CeSZ ceramics sintered at different temperatures were investigated, using XRD, SEM, impedance spectroscopy, as well as XPS, with the help of which an unambiguous and highly accurate determination of the chemical state of the atoms involved, along with a compositional analysis, was possible. 

As expected, there was a significant increase in the grain ionic conductivity with the sintering temperature. This increase was accompanied by a decrease in the porosity of the samples, an increase in the grain size, and a transformation from the rhombohedral phase to the cubic phase. The transformation from the rhombohedral to the cubic phase was detected via XRD, and confirmed indirectly via XPS, based on the analysis of the Zr-3d, Sc-2p, and O-1s, along with the Ce-3d states. 

A detailed analysis of the Zr-3d states revealed a greater amount of Zr–OH bonds on the surface of the ceramic sintered at the lower temperature of 1100 °C than on the ceramic sintered at the higher temperature of 1400 °C. The existence of these Zr–OH bonds was confirmed using FTIR spectroscopy. A detailed XPS analysis revealed an increase in the number of oxygen vacancies in the ceramic sintered at the higher temperature. This allowed us to discuss, as well as the structural factors, the essential role OH^–^ ions play in the transport properties of 10Sc1CeSZ ceramics sintered at different temperatures: OH^–^ ions influence the number of active oxygen vacancies. 

## Figures and Tables

**Figure 1 materials-16-05504-f001:**
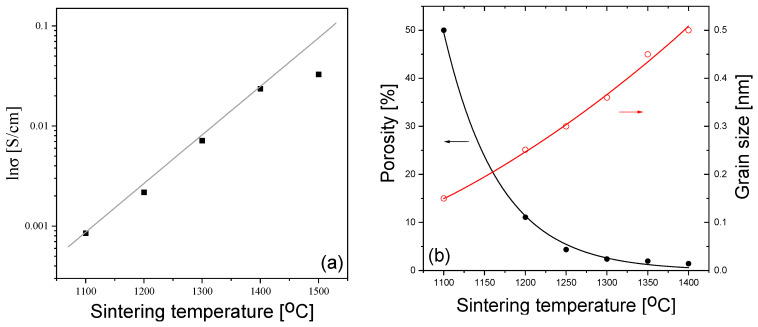
(**a**) The dependence of the grain conductivity of 10Sc1CeSZ ceramics on the sintering temperature, measured at 500 °C. (**b**) The dependence of the ceramic porosity and grain sizes on the Ts.

**Figure 2 materials-16-05504-f002:**
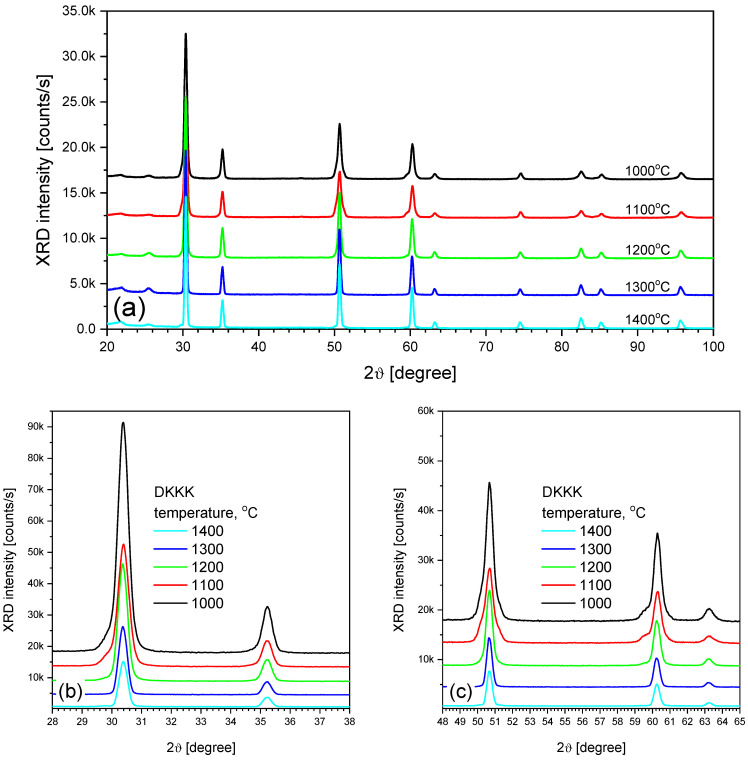
XRD patterns of ceramic samples sintered at different temperatures: (**a**) full, and (**b**,**c**) zoomed ranges.

**Figure 3 materials-16-05504-f003:**
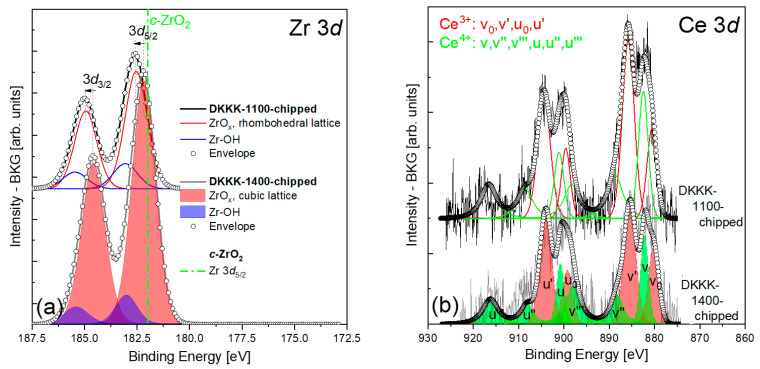
(**a**) Zr 3d and (**b**) Ce 3d spectra of the DKKK-1100 and DKKK-1400 samples. The position of the Zr 3d_5/2_ line at 182 eV corresponds to the c-ZrO_2_ phase, according to [22], and is marked by the green dash–dot line in (**a**). The assignment of the different peaks u and v (and their derivatives) is related to the different initial and final states of Ce^3+^ and Ce^4+^ ions in the Ce 3d core level in the XPS spectra (this notation is adopted from Table 2 in Ref. [35]).

**Figure 4 materials-16-05504-f004:**
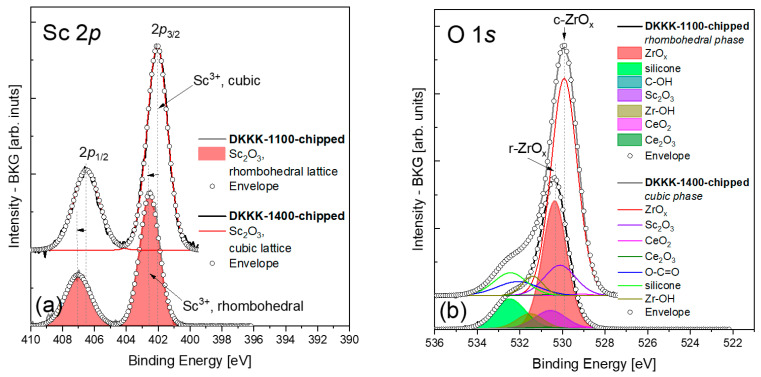
(**a**) Sc 2p and (**b**) O 1s spectra of the DKKK-1100 (**top**) and DKKK-1400 (**bottom**) samples.

**Table 1 materials-16-05504-t001:** The atomic percentages of elements obtained via the deconvolution procedure.

	ZrO_x_, at%		Zr-OH, at%		Ce_2_O_3_, at%		CeO_2_, at%		Sc_2_O_3_, at%		C-C, at%
Considered Line	Zr 3d	O 1s	Zr 3d	O 1s	Ce 3d	O 1s	Ce 3d	O 1s	Sc 2p	O 1s	C 1s
DKKK-1100 (R)	21.91	40.06	5.20	5.19	0.44	0.65	0.31	0.62	5.09	7.66	12.77
DKKK-1400 (C)	26.01	39.05	2.80	2.79	0.17	0.27	0.11	0.22	4.42	6.58	17.49

## Data Availability

Data are available from the corresponding author upon reasonable request.

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
