# Peer review of "X-ray Photoelectron Spectroscopy Analysis of Scandia-Ceria-Stabilized Zirconia Composites with Different Transport Properties"

_materials, 2023, doi:10.3390/ma16165504_

Round 1
Reviewer 1 Report
Title: XPS analysis as a tool to assess factors affecting deterioration of transport properties of Scandia-Ceria Stabilized Zirconia composites
Overall, the presentation style of article is fine. However, the authors should consider following comments before its acceptance for publication. The publication can be acceptable after corrections. These comments will help the article to attract by readers.
1. The title seems good, however the abstract will be good after adding few concept of findings in it. Please use full standard form of any term in abstract.
2. Please mention the short form of any term before using it.
3. Introduction section must be written on more quality way, i.e., more up-to-date references addressed. Overall, the structure of introduction is fine. Please update latest references such as Catalysts 13 (2), 409, Scientific Reports 12 (1), 14023.
4. The novelty of the work must be clearly addressed and discussed.
5. How the author can claims the a greater amount of -OH hydroxide group on the surface of ceramic without analysing by FTIR or any other technique.
6. I did not find the results of SEM.
7. Please add FTIR and EDX results for better understanding.
8. Please provide space between number and units. Please revise your paper accordingly since some issue occurs on several spots in the paper.(line number 91) etc.
9. Please check the abbreviations of terms throughout the article. All should be consistent.
10. Need to check all the typo errors as well as improve the quality of the English language.
11. Revise the abstract and conclusion based on the new results.
Need to improve.
Reviewer 2 Report
The article should be rewritten in a better style (English usage and use of technical terms and units). Moroever, all Figures need to be redrawn.
Figure 1: inconsistent use of two names for sintering temperature and please do not name a temperature given in °C as "t" as this is not IUPAC conform.
Figure 2: please use the same order for the temperatures in 2 a, b and c. In a the upper temperature is 1400 °C and in b and c it is vice versa.
Figures 3 and 4: too much information in one picture. Either make them bigger or separate. Also mention what the symbols are used for (v, v`...)
Table 1 needs to be shifted according to the first mention in the text, e.g., put it down. For Figure 3 it is opposite, put it higher to the text.
Comments on the style and contents:
- Title is too long "a tool to assess factors"
- abstract: some facts are just standard of knowledge (increasing T accompanied with decrease in porosity - is not new); use of terms like "XPS method" inappropriate - similar valid for the overall text; last sentence is not correct
- introduction: text has to be overworked; first sentence: "The subject of these studies"; too often "has"; use of articles: mainly missing "the" before nouns or wrong use instead of "a"; line 49: "ScZr`" minus?; use of °C writing style should be changed throughout paper; same for mol%; use of "Ts" instead full term is not appropriate, just write it as word, line 69: do not use the term "laws", the meaning of law and a simple correlation or relation is different;
- materials and methods: change format line 90; remove term "XPS method", just write by XPS; change sentence line 97 and 98 in a correct language style; line 104 remove "impedance methods", just write impedance spectroscopy; line 120 just "procedure for XP spectra"; rewrite paragraph 121-130 as it does not play a role, what was at the beginning, just say something like "due to surface treatment changes may be obsereverd, and therefore you did not clean it..."; line 129: it is not clear what you mean by " inside the samples";
- results and discussion: use of °C wrong; lines 139-144: not new; lines 144-146: several times "another"; line 154: "XRD data", just write from XRD; line 160: remove "which should be taken into account". It is self-explaining that one should do it.; put table somewhere between line 170 and 180; lines 175 to 180: why not calibrating it also according to literature when you compare this?; line 201: check sentence; line 205: "XRD analysis", change it also the followings word constructions with "XPS data or XPS analysis", this is clear from the subject of the paper that you used for analysis XPS and XRD; ; lines 251 and following: I do not confirm with the OH interpretation and description;
please remove the term water gas!!!!, it is gaseous adsorbed water oder adsorbed water vapour! water gas is a chemical compound composition and does mean something totally different; why should a higher porosity provoke higher OH-concentrations? did you ever do a TPD? such strong OH bonds are unlikely?; lines 262-298: can you give another prove for Ce3+, maybe ESR?;
The whole text needs to be carefully checked again. I did not put all missing "a", the wrong "the" here.
There is no clear evidence for a connection between your title and the text. What do you mean by transport properties and which properties deteriorate?
conclusions: the style of the abstract is found here again. See hints above.
see above
Round 2
Reviewer 2 Report
I agree with your changes, still I am not convinced by your FT-IR however I know abaout the difficulties of assigning -OH in the broad bands of Zr-OH. You should maybe not overinterprete those bands as they are not really at 3760 cm-1. But as discussion basis, it is ok.